# GenomeBits Characterization of MPXV

**DOI:** 10.3390/genes13122223

**Published:** 2022-11-27

**Authors:** Enrique Canessa

**Affiliations:** International Centre for Theoretical Physics—ICTP, 34151 Trieste, Italy; canessae@ictp.it

**Keywords:** monkeypox virus, MPXV surveillance, genome sequence, GenomeBits

## Abstract

Genome sequences of the monkeypox virus (MPXV) causing the current outbreak are being reported from an increasing number of countries. We present a letter-to-numerical sequence study via GenomeBits signal mapping in order to characterize the evolution of the MPXV via simple statistical analysis. Histograms, empirical and theoretical cumulative distribution curves and the resulting scatter plots for the base nucleotides A and C versus their complementary base nucleotides T and G are discussed. GenomeBits may help the surveillance of emergent infectious diseases.

## 1. Introduction

According to WHO, the monkeypox virus (MPXV) poses new threats to humanity [1]. As surveillance expands in non-endemic countries, the cases of monkeypox being identified continue to rise with the risk of becoming an outbreak. Proper precautions and steps are necessary to avoid another global pandemic in 2022–2023.

MPXV is an infectious disease caused by the Zoonotic Orthopoxvirus closely related to cowpox and smallpox viruses—first identified in a monkey in Denmark at the end of the 1950s. Although it is mainly transmitted by monkeys and rodents, the human-to-human spread is also prevalent since the first case was recorded in central Africa in 1970. As of today, individuals infected by the MPXV are reported across continents. MPXV is transmitted through close physical contact with an infected person, animal or object, and its principal long-term effect is the presence of human skin lesions.

Understanding this pathogen better is still a global challenge for scientific research. A global network of permanent bioinformatics surveillance, with the capacity to rapidly deploy genomic tools and statistical studies for the MPXV characterization of the evolution of lineages and mutations, is still much needed. There are hundreds of complete genome sequence samples reported from confirmed cases from around the world [2,3,4], which indicate a close match to exported cases from Africa found in 2017 [5].

In this paper, we present a novel method for characterizing the MPXV genome from available data sequences. We apply the novel GenomeBits statistical algorithm, whose salient feature is to map the nucleotide bases α=A,C,T,G (as observed along a genome sequence) into a finite alternating (±) sum series of distributed terms of binary (0,1) indicators [6]. We show how GenomeBits allow infectious MPXV surveillance by tracking and analyzing viral sequences in relation to their evolution status. In particular, we derive scatter plots for the nucleotides A-T and C-G to uncover correlations between MPXV sequences of complementary nucleotide bases.

## 2. Method

The new quantitative method presented for the characterization of distinctive patterns of complete genome sequences of length *N* considers
(1)$$E_{\alpha ,N}\left(X\right)=\sum_{k=1}^{N}{\left(-1\right)}^{k-1}X_{\alpha ,k}\;,$$
where the independently distributed terms $X_{k,\alpha }$ are associated with (0,1) according to their position along the sequences, satisfying the following relation $X_{\alpha ,k=N}=\left|E_{\alpha ,N}\left(X\right)-E_{\alpha ,N-1}\left(X\right)\right|$. The arithmetic progression carries positive and negative signs ${\left(-1\right)}^{k-1}$ depending on the nucleotide base position *k*.

This mapping into four binary projections of genome sequences follows previous studies on the three-base periodicity characteristic of protein-coding DNA sequences [7]. However, as a principal difference from other binary representations (see also, e.g., Refs [8,9]), in Equation (Equation 1), the terms in the sums change sign. If a term $X_{\alpha ,k}$ in the sum series is positive at a given *k*, then the next $X_{\alpha ,k+1}$ term is negative, and vice versa.

The binary indicator for the sequences is chosen sequentially, starting with +1. This is illustrated in Table 1, where a mapping example for converting the brief genome fragment TTTTAGTACATTAAT (consisting of 15 nucleotides) into the alternating binary array via Equation (Equation 1) is given. As illustrated in Table 1, the positive and negative terms in the sums of Equation (Equation 1) partly cancel out, allowing the sum series to diverge from zero rapidly and to become a non-Cauchy sequence type [6].

There is a user-friendly Graphics User Interface (GUI) for the GenomeBits signal analysis method of genome sequences in FASTA format. The GUI can be downloaded freely from GitHub (see https://github.com/canessae/GenomeBits/, accessed on 26 November 2022) [10]. It runs under Linux Ubuntu OS. For large genomic data, say, of the order of *N* = 197,204 nucleotide bases as, for example, for ON563414.3 monkeypox virus isolate MPXV_USA_2022_MA001 complete genome [3], the one used in the present study, the GUI requires little processing time and discards uncompleted sequences containing codification errors (usually denoted with “NNNNN” letters).

## 3. Discussion

Next, we show some interesting statistical imprints of the intrinsic gene organization at the level of nucleotides from the above formula. In Figure 1, the results obtained via Equation (Equation 1) for the binary sequences of each A, C, G and T nucleotides of the MPXV reported from the USA are shown. For comparison, the curves Figure 1a on the top display results for nucleotides A and T of the strand and on the bottom Figure 1b are the nucleotides C and G (complementary to those of the opposite strand according to the pairing rules A-T and C-G of DNA). It is interesting to note how in the figure there are regions where the curves seem to mirror each other. This peculiar behavior reveals coding regions of correspondence between single-nucleotide structures.

Moreover, the fact that for α=(A), the total number of zeros is 66.4% of the whole sequence and the total number of ones is 33.6% is also interesting. For α=(C), these correspond to 83.5% and 16.5%, respectively. For α=(G), one finds 83.5% and 16.5% and, similarly, for α=(T), the values become 66.6% and 33.4%.

A first step in assessing the probability distribution of datasets is usually obtained by computing a basic histogram (i.e., counting the number of data points in a given bin). A standard cumulative, normalized histogram as a step function in order to visualize the empirical cumulative distribution function (CDF) for a sample is also useful [11]. In this case, one gets the probability of, and observation from, the sample not exceeding a particular *x*-value at a fixed *y*-value. However, our goal here is to compare two different datasets or, alternatively, two different histograms of equally sized bins.

To achieve this, a pair of histograms above and to the right of a scatter plot of such two datasets can help to visualize and highlight correlations at certain colored contact points. Scatter plots with marginal distributions of this kind can be easily made using Python programming through the built-in Matplotlib’s routines: scatter(x,y), hist(x), gridspec and invert_xaxis (see, e.g., Ref. [12]). In Figure 2 and Figure 3, we illustrate the (blue and green bar-graph) histograms, the empirical (red full line) and theoretical (red dotted lines) CDF curves, and the resulting scatter plots for the nucleotides A and its complementary T and C and its complementary G plotted in Figure 1. We have used 50 bins, and the fits (blue line) have been calculated via an expected Gaussian distribution.

In Figure 2 and Figure 3, we have drawn scatter diagrams in an attempt to obtain correlations and anomalies between GenomeBits sequences of complementary nucleotide bases. The clearer colors of the hexagon markers add a new dimension to these plots. These show how the mapped binary data is stratified along the sequences and which areas show somehow correlated points that fall along complex, rounded shapes. The markers are unequally distributed in the scatter plot and far from forming a straight line. The patterns found that summarize the underlying data do not occur by random chance.

From these results, we believe GenomeBits may help to shed light on the surveillance behind infectious diseases. By GenomeBits, we have also uncovered distinctive signals for the intrinsic gene insights in the coronavirus genome sequences for past variants of concern: Alpha, Beta and Gamma, and the variants of interest: Epsilon and Eta [6]. A study of GenomeBits into the most recent delta and omicron variants of the coronavirus pathogen can be found in Ref. [13] (see also [14]). The GenomeBits method focuses on single-nucleotide structures and may also provide useful information to study the evolution of the MPXV outbreak via simple statistical analysis.

## 4. Conclusions

Our main contributions here are the novel and exciting scatter plot patterns between A-T and G-C of MPXV sequences as extracted from the GenomeBits algorithm. Histograms, empirical and theoretical cumulative distribution curves, and the resulting scatter plots for pairs of base nucleotides have been presented. These statistical imprints of genome sequences by data mining can be a valuable tool to identify insightful patterns and convey cluster anomalies present within two datasets of nucleotides. Scatter plots, in general, such as those in Figure 2 and Figure 3, offer a simple and fast form of data visualization (with little cognitive effort) of a pair of variables when plotted on a plane as a set of colored points. In the presence of high-dimensional datasets, scatter plots of all possible combinations of two variables are usually combined to build a scatter plot matrix and so extract patterns present in these datasets. The event patterns allow for visual cues and comparisons, and their structures are expected to be different along mutations of the virus based on measurements of their genome sequences.

## Figures and Tables

**Figure 1 genes-13-02223-f001:**
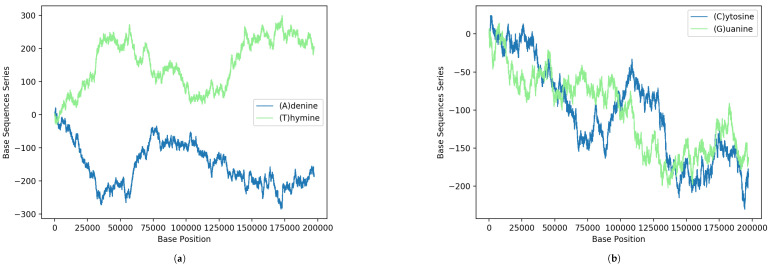
GenomeBits imprints of the genome sequence monkeypox virus isolated in the United States MPXV_USA_2022_MA001 (GenBank ID ON563414.3): (**a**) Results according to the pairing rules A-T and (**b**) pairing rules C-G.

**Figure 2 genes-13-02223-f002:**
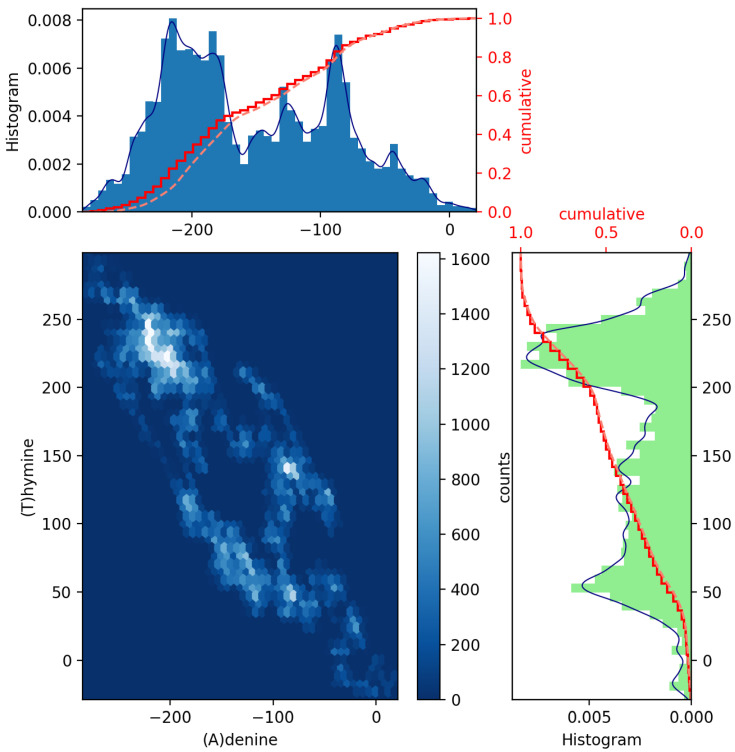
Scatter plot of GenomeBits (A)denine and (T)hymine curves with histograms for the monkeypox virus (GenBank ID ON563414.3).

**Figure 3 genes-13-02223-f003:**
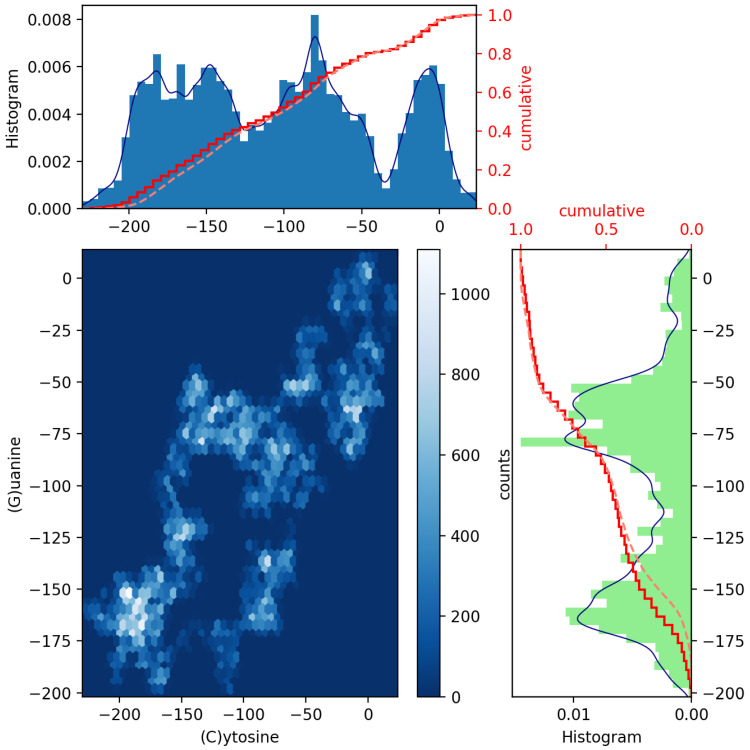
Scatter plot of GenomeBits (C)ytosine and (G)uanine curves with histograms for the monkeypox virus (GenBank ID ON563414.3).

**Table 1 genes-13-02223-t001:** Genome sequence-to-GenomeBits mapping example via Equation (Equation 1) for *N* = 15.

Base Position *k*	1	2	3	4	5	6	7	8	9	10	11	12	13	14	15	
**Sequence (String)**	**T**	**T**	**T**	**T**	**A**	**G**	**T**	**A**	**C**	**A**	**T**	**T**	**A**	**A**	**T**	**GenomeBits Sums**
${\left(-1\right)}^{k-1}X_{A,k}$	0	0	0	0	+1	0	0	−1	0	−1	0	0	+1	−1	0	−1
${\left(-1\right)}^{k-1}X_{C,k}$	0	0	0	0	0	0	0	0	+1	0	0	0	0	0	0	+1
${\left(-1\right)}^{k-1}X_{G,k}$	0	0	0	0	0	−1	0	0	0	0	0	0	0	0	0	−1
${\left(-1\right)}^{k-1}X_{T,k}$	+1	−1	+1	−1	0	0	+1	0	0	0	+1	−1	0	0	+1	+2

Letter sequence-to-numerical GenomeBits signal mapping. Single letters represent nucleotide codes: (A)denine,(C)ytosine, (G)uanine and (T)hymine. Signs are assigned sequentially with + at base position *k* = 1.

## Data Availability

All custom code used in the manuscript is available at https://github.com/canessae/GenomeBits/, accessed on 24 November 2022.

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
