# Peer review of "GenomeBits Characterization of MPXV"

_genes, 2022, doi:10.3390/genes13122223_

Round 1
Reviewer 1 Report
The article titled "GenomeBits characterization of MPXV" by E. Canessa is a fairly novel concept and an interesting work in viral genomics. While the author did a great job in designing the study and the presentation of results and discussion, this reviewer believes that the Introduction section is too short and therefore needs a revision for a better reading and understanding.
The following are minor comments for the author to consider:
P2-Line 35: As illustrated in Table 1....
P2-Line 40: cab be downloaded freely from the Github (provide the download link).
Can Introduction be numbered '0' as done in the manuscript?
Author Response
I would like to thank Referee #1 for considering the paper a great job in designing the study and the
presentation of results and discussion. I have revised and rewritten the Introduction section as suggested for
a better reading and understanding.
Reviewer 2 Report
The genome bits algorithm is an exciting initiative in characterizing the MPXV genome. The scatter plot patterns between A-T and G-C sequences are novel and exciting. However, more validation of the concept and the statistical practices are required before considering the general utility of this method in infectious disease surveillance. In the MS more material on the introduction and discussion is needed. It will be more pertinent to have a more lucid explanation regarding the statistical imprints observed in the study. Overall a very interesting and novel concept.
Author Response
I also thank Referee #2 for considering the GenomeBits algorithm as an exciting method to characterizing
the MPXV genome. I have revised and rewritten the Introduction and added a new Conclusion section as
suggested to give more comprehensive details regarding the statistical imprints of GenomeBits.